# How Can Adolescents Benefit from the Use of Social Networks? The iGeneration on Instagram

**DOI:** 10.3390/ijerph17196952

**Published:** 2020-09-23

**Authors:** Sabrina Cipolletta, Clelia Malighetti, Chiara Cenedese, Andrea Spoto

**Affiliations:** 1Department of General Psychology, University of Padua, 35100 Padua, Italy; cenedese.chiara92@gmail.com (C.C.); andrea.spoto@unipd.it (A.S.); 2Department of Psychology, Catholic University of the Sacred Heart, 20019 Milan, Italy; clelia.malighetti@gmail.com

**Keywords:** adolescence, Instagram, Internet, repertory grids, social media, social networks

## Abstract

In the last few years, Instagram has been a topic of much contention, as it has been shown to be associated with both risks and benefits for young users. This study explores the influence of the use of Instagram on adolescents’ constructions of self and interpersonal experience. Forty Italian adolescents aged between 11 and 16 years were interviewed and completed repertory grids. The results showed that the adolescents’ self-construction and distance from others were mostly influenced by receiving, or not receiving, positive feedback, rather than by using Instagram itself. Specifically, there was an increase in self-acceptance and social desirability after receiving a “like” and an increase in social isolation after receiving no “likes”. The regression model also showed a decrease in self-acceptance on Instagram in the case of female adolescents, and in participants who edited photos. These findings are useful for understanding the constant need for approval adolescents require today and could be used as a guiding tool for future studies and intervention policies. The present study offers an innovative methodology that refers to the relevant dimensions of adolescents’ self-construction rather than investigating the more general relationship between personality traits and social networks’ use.

## 1. Introduction

The past few years have been marked by the incredible growth and popularity of social networking sites (SNSs), such as Instagram and Facebook [1]. These are defined as “websites, which make it possible to form online communities and share user created content” [2] (p. 216). Social networking sites enable users to create public, or semi-public, profiles, and to connect that profile to others to form a personal network [3]. The access to these platforms via ubiquitous technologies has enabled people to document nearly every aspect of their everyday lives, to present themselves and manage their social relationships online, to communicate with friends who are part of their social network and, from every part of the world, to be always updated on what is happening, to interact with other cultures without time limits, to access health-promoting information and, also, to facilitate scholastic research and study [4,5,6,7]. Furthermore, the immediate, low cost, private, and hidden communication provided by social networks [8] has helped to make SNSs a common online destination for adolescents [9,10]. People who connect the most are the so-called “iGeneration” or iGen [11], the post-Millennial generation who were born after 1995, and who have grown up using mobile devices, social media, and constant connectivity [12].

A nationally representative survey [13] of 790 American teenagers showed that nearly all teens aged 13 to 17 (94%) use social media platforms and more than half (56%) go online several times a day. This survey also showed that American teens have shifted their favored social media platforms and are now most likely to use Instagram (76%) and Snapchat (75%). At present, little information is available on the social activities of this age group in other parts of the world. A recent report [14] on the use of technology among young European teenagers involved young people aged 16 to 24 years and revealed that almost 9 in 10 people in the EU participated in social networks (88%).

According to data from the National Adolescence Observatory [15], 94% of Italian teenagers use the Internet to talk with friends, and more than half (54%) use it to check their social profile. Many adolescents begin and end their day by checking SNS posts. The creation and maintenance of friendship networks is considered an important and developmentally significant process during adolescence [16,17,18]. During this life stage, the peer group often assumes a key importance and displaces parental relationships as the principal source of social support [19].

One of the most popular applications, especially among teens, is Instagram [20]. Since its beginnings in 2010, it has attracted more than 500 million active users, who upload around 95 million photos a day. A recent study revealed that the two key reasons for using Instagram are self-expression and social interaction, suggesting that Instagram users utilize pictures to present their actual and ideal selves, as well as to maintain social relationships [21]. However, unlike Facebook, Instagram is used mainly as a method of self-promotion and does not focus so much on social relationships [22,23].

A crucial feature of Instagram is the ability to like an image, allowing a straightforward measure of peer endorsement, and the possibility of becoming a potential source of peer influence [24]. Sherman and colleagues [25] found that adolescents were more likely to like photos that had received more “likes” from peers, even if they were strangers. The positive mental health benefits of using Instagram, such as new opportunities for sociability and self-expression, have been reported [8], but Jackson and Luchner [26] showed that a person’s personality influences how they present themselves on social networks and their emotional responses to feedback. Research is moving away from the investigation of the effects of the use of SNSs to the moderating variables of these effects.

In this context, the use and consequences of SNSs has been a topic of much contention, as they have been shown to be associated with both risks and benefits for young users [27]. They allow teens to accomplish online many of the tasks that are important to them offline, such as entertainment and information-seeking, staying connected with friends and family, facilitating intragroup–intergroup relationships, making new friends, sharing pictures, and exchanging ideas [3,17,28,29]. In particular, there is evidence [9] that SNSs enhance the capacity for (online) socializing and reduce feelings of loneliness [30,31]. Increased social networking opportunities raise self-esteem, feelings of belonging, and the chance for online self-disclosure, which may then indirectly impact upon feelings of wellbeing [32]. In turn, self-disclosing and associated positive feedback can enhance perceptions of community integration and social support [33,34,35]. Valkenburg et al. [32] found that adolescents use social networks to find out how others react to them, to overcome shyness, and to facilitate relationship formation. Interestingly, those who feel less secure in face-to-face contexts report a preference for Internet interactions [32,36,37].

On the contrary, some studies have highlighted online risks, such as depression due to negative social debate [28], body dissatisfaction, the drive for slimness [38,39,40] and orthorexia nervosa [41], cyber-bullying, social isolation, and exploitation [30,41,42,43]. Specifically, adolescents spend a considerable portion of their daily lives on SNSs, leading them to neglect academic, physical, and face-to-face social activities [28,44,45].

In light of this literature, it remains unclear whether, and to what extent, SNSs use leads to positive or negative consequences for the most avid participants, that is, adolescents [46]. Moreover, research involving this population is still sparse because the majority of the studies on social media involve college students [47]. Additionally, a recent review [48] pointed out that previous studies have mainly focused on the positive versus negative impacts of SNSs on adolescents’ identity development and social relationship formation, whereas it seems essential to further investigate the mechanisms by which different aspects of a SNS may interact with the varied dimensions of adolescents’ identity.

Departing from the assumption that adolescents engage in selective self-presentation online [49], and feedback from these presentations may influence their self-concept [50], the current study explored the impact of different and discrete activities (e.g., being on Instagram and receiving a “like”) on varied dimensions of the adolescents’ construction of self and their interpersonal experience: looking-glass self, self-acceptance, social desirability, change, and social isolation. These dimensions have been defined within a constructivist framework and, in particular, refer to personal construct theory (PCT) [51]. The concept of “looking-glass self” was introduced by Cooley [52] who considered the self-concept to be the result of the perception that individuals have of the perceptions that their significant others have of them. Self-acceptance refers to the discrepancy between the actual self and the ideal self, and social desirability to the distance between present self and a person one likes [53]. Change refers to the distance between the actual self and past self [54]. Social isolation, referring to the feeling of estrangement experienced in relation to other people, is given by the discrepancy between self-perception and the perception of others, which indicates that a person sees him/herself as being unlike anybody s/he knows, thus, representing him/herself as being alone [55]. The aforementioned dimensions have never been explored in relation to SNSs use.

The aim of the present study was precisely to understand how the distance of actual self from other selves (past and ideal) and other people changes when adolescents evaluate themselves on Instagram and, specifically, after receiving or not receiving a “like”. The following hypotheses were derived from the results of previous studies that explore similar dimensions:Adolescents’ construction of self-adheres less to their construction of others’ construction of them (looking-glass self) when they evaluate themselves on Instagram than when they evaluate themselves offline, because on Instagram they find new opportunities for self-expression [55,56].Self-acceptance and social desirability are higher on Instagram and after receiving a “like” and lower after receiving no “like” than when adolescents evaluate themselves offline, as suggested by the results of previous studies [57,58,59] on the impact of Instagram on self-esteem.The perceived change on Instagram is higher than the change perceived when adolescents think of themselves offline due to online self-disclosure and to the additional opportunities offered by SNSs for social comparison across groups [60,61].Social isolation is lower on Instagram and after receiving a “like” than offline because Instagram offers the opportunity to connect with other people [55,56].Social isolation is higher after receiving no like than when the adolescent is offline because, as previous studies [30,62] suggest, not receiving any positive feedback may lead to a greater sense of loneliness.

## 2. Materials and Methods

### 2.1. Participants

Forty adolescents aged between 11 and 16 years (mean age = 14.3, SD = 1.2) participated in the study. The sample consisted of 23 girls (mean age = 14.1) and 17 boys (mean age = 14.6), attending the last two years of secondary school and the first two years of high school and living in the same geographic area of Northern Italy (i.e., Veneto region). Participants were all Instagram users and were recruited through the “snowball sampling” method, a non-random sampling method in which the individuals selected to be studied recruit new participants from among their circle of acquaintances [63]. One of the researchers approached the first participants among her personal acquaintances and reached the other participants through them.

After obtaining approval from the Ethics Committee of Psychology Research of the University of Padova, parents and adolescents gave their written informed consent with regard to participation in the study.

### 2.2. Measures

Socio-biographic characteristics (sex, age, nationality, and education) and the exploration of the participants’ use of Instagram were obtained through a specifically designed interview. In particular, for each participant, the interview investigated the frequency of the use of the Internet and specifically of Instagram (asking for an overall estimate of time spent on social media throughout the day); the reasons that led him/her to register; if he/she had a public or private profile; the activities that he/she performed on Instagram and, therefore, why he/she uses it; if he/she publishes photos; if he/she changes them before posting them; if he/she comments on the photos of others; and if he/she adds “likes” to the photos of others; the number of followers; and whether or not his/her parents knew that he/she enjoyed Instagram.

After the administration of the interview, as detailed above, the repertory grid technique was used to investigate the adolescents’ self-construction as Instagram users. The entire procedure required about one-and-a-half hours to complete.

The repertory grid technique [51,64] is a semi-structured interview underpinned by PCT, which consists of elements and constructs. Elements consist of significant people for the person completing the grid and aspects of the self. For the repertory grid used in the present study, elements included self on Instagram, offline self, ideal self, future self, past self (before using Instagram), how others see me, the person I like, the person I do not like, mum, dad, my best friend, my body now, my body in the past, self after the “like”, and self after receiving no “like”. The constructs were elicited through the triadic method, presenting sets of three elements and asking, for each triad, for a way in which two of the elements were similar and, thereby, different from the third. The participant was then asked to rate the elements on each construct on a −3/+3 point scale, which represented the bipolarity of the constructs. An example of a repertory grid is presented in Appendix A.

The conventional criteria of validity and reliability in evaluating the psychometric properties of repertory grid data must be reconsidered in the light of the fundamental postulate of the PCT [51] that individuals are, as scientists, involved in the anticipation of their worlds through the formulation and testing out of hypotheses, or constructions of events, and revision of these if they are invalidated. The repertory grid is a tool used to capture construction systems and it has no specific content that espouses to measure a trait (as a questionnaire is designed to do). Thereby, the validity is linked to the possibility of revealing patterns and relationships in certain kinds of data, whether it does so effectively or not. In terms of reliability, the grid should be evaluated not in terms of whether it has a “high” or “low” consistency, but whether or not it is an instrument which enables us to effectively inquire into the way in which people maintain or alter their construing events [64].

### 2.3. Data Analysis

Repertory grids were analyzed using the Idiogrid software program [65] in order to calculate a range of indices [52,60] as follows:Looking-glass self, the Euclidean distances “offline self–how others see me”, “self on Instagram–how others see me”, and “self after like–how others see me”;Change, the distances “offline self–past self”, “self on Instagram–past self”, and “self after like–past self”;Self-acceptance, the distances “offline self–ideal self”, “self on Instagram–ideal self”, and “self after like–ideal self”, with the lesser the distance, the higher the self-acceptance;Social desirability, the distances “offline self–a person I like” and “self on Instagram–a person I like”, “self after like–a person I like”, with the lesser the distance, the higher the desirability;Social isolation, the mean of the distances “offline self–father”, “offline self–mother”, and “offline self–best friend”, the mean of the distances “self on Instagram-father”, “self on Instagram–mother”, and “self on Instagram–best friend”, and of “self after like–father”, “self after like–mother”, and “self after like–best friend”.

Before performing the descriptive and inferential statistics, the data collected through repertory grids were pre-processed in order to ensure that all data matrices were complete. No missing data were observed for either repertory grids or interview sections. We also checked for particular response patterns (such as central tendency bias) and we did not notice any issue. Therefore, all the 40 recruited participants were included in the analysis.

Different 2 × 2-mixed model ANOVA tests were used to evaluate the differences between the scores of the indices obtained by the grids. In each model, we tested the effect of gender as between factors, and of the specific indices involved, as within factors. Moreover, models of linear regression were estimated using the list wise procedure in order to evaluate the role of individual level effects like gender, the person’s public/private profile, age of the user, number of followers, usage frequency, and the eventual editing of the posted photos.

Given the analysis plan, we estimated the sample size by means of the software G-Power 3.1.9.2 [66]. We referred to the possibility of detecting even a low to moderate effect (partial η^2^ > 10) with a power of 0.95 with α = 0.05 and a relatively low correlation among repeated measures (<0.40). The estimated sample size was actually 40.

## 3. Results

### 3.1. Adolescents’ Use of Social Networks

The analysis of the frequency of the use of the Internet, specifically of Instagram, showed that all the participants accessed the Internet (100%) and chatted with their friends via Whatsapp (95%) every day via mobile devices. In addition to Instagram, 27.5% of the participants had joined Facebook (29.4% of the males and 26.1% of the females) but they hardly used it, seeing it as “too boring” and for “older people”. Instead, the majority considered Instagram to be “funnier and better”, with the exception of two 15-year-olds who preferred Facebook because it was “more varied” and “because on Instagram there are only photos”. All the girls and 15 out of 17 boys used Instagram every day with girls spending an average of 3.21 h a day on Instagram and boys 1.6 h. With regard to the age of registration, the interviews revealed a mean age of 13 years for males and 12 for females; for this sample, 47.6% of males (i.e., 8 out of 17 boys) and 69.6% of girls (16 girls out of 23) had signed up before the age of 13 (the minimum age for registration).

The reasons that led participants to register were mainly to align themselves with others, because everybody used Instagram, and to share their photos and videos. All the participants had only one profile on Instagram, four out of 17 boys (23.5%) had a public profile; the remaining 13 had a private profile; 3 of 23 girls (13.1%) had a public profile, while 86.9% had a private profile.

The reasons why participants used Instagram were varied, such as for looking at the photos and videos of others, and to see what friends were doing, to follow celebrities, to chat and stay in touch with friends, to share what they are doing, and to publish their own videos and photos.

With regard to the participants’ use of filters to edit photos before posting them on Instagram, 70.6% of the males declared that they did not edit their photos, the remaining 29.4% (5 boys out of 17) admitted to editing photos by applying the filters which are available on Instagram; for girls, however, the situation was different, in that 6 out of 23 girls (26.1%) declared that they did not modify them, while the remaining 73.9% (17 out of 23) did edit their photos, mostly using the available filters.

Regarding the number of followers, the analysis showed differences between females and males. Males had a minimum of 50 followers and a maximum of 1200 (with an average of 398 followers); while females had a minimum of 130 followers and a maximum of 2066 (raising the average to 846.6). The interview also explored what participants thought about the number of followers and, in relation to this, what others may think. Males who had fewer than 300 followers were aware that others would think that they had “too few” and justified it on the basis of limited use of Instagram; they considered that having between 500 and 650 followers was normal. The girls considered that fewer than 200 followers was “few” and the number of followers ranging from 348 to 2066 was “normal”. The girls believed that others might think that having “few” followers was for “losers”, while more than 800 followers made you popular, famous, and well-known.

Finally, we asked if the parents knew that they had joined Instagram. All respondents answered affirmatively, except for two sisters who had kept it hidden from their father, while their mother knew about it.

### 3.2. Repertory Grids

All the values of the main effects observed within the factors of each ANOVA test with respect to the indices investigated in the research are shown in Table 1.

With respect to the looking-glass self, a significant main effect of the within factor was observed, indicating a significant difference between the two distances “offline self–how others see me” and “self on Instagram–how others see me” with the former being lower than the latter. Moreover, the same main effect was found with respect to the difference between “offline self–how others see me” and “self after like–how others see me”. Additionally, in this case the former variable was lower than the latter. The comparison between the average distances between “offline self–past self” and “self on Instagram–past self” provides the perception of change related to the use of Instagram. The distance between “self on Instagram” and “past self” was significantly higher than the distance between “offline self” and “past self”. As regards self-acceptance, Table 1 shows a significant effect in terms of the within factor in the distances between the “ideal self” and the “self after a like” and the “self after no like”, while there were no statistically significant differences in comparing the difference “ideal self–self on Instagram” with the distance “ideal self–present self”. Concerning social isolation, the results showed no significant difference in the distance “self on Instagram–others”, compared to the average of “offline self–others” distance, but the distance “self after no like–others” was higher than the distance “offline self–others”.

With regard to social desirability, there were no significant differences in terms of the distance between “offline self–the person I like” and “self on Instagram–the person I like”. On the contrary, Table 1 shows a lesser distance between “self after like” and “person I like” than “offline self–person I like” and a greater distance between “self after no like” and “person I like” than “offline self–person I like”. The main results of the comparisons between the indices calculated on repertory grids are summarized in Figure 1. In general, no effect of the gender factor was observed.

With respect to the regression models, we observed that the difference between the “self” on Instagram and “ideal self” was significantly predicted (F2, 37 = 3.105; *p* = 0.05, R2 = 0.144) by the variables, gender (β = 0.359; t38 = 2.114; *p* < 0.05), and editing of the posted photos (β = −0.359; t38 = −2.119; *p* < 0.05). This result indicates that the difference tends to increase in female adolescents and if the photos posted are edited. No significant regression model was obtained either for the difference between the variables “self after like” and “ideal self”, or for the difference between the variables “self after no like” and “ideal self”, indicating that none of the predictors considered have an effect on these variables. On the contrary, the difference between the variables “self after like” and “person I like” is significantly explained (F3, 36 = 2.964; *p* < 0.05, R2 = 0.198) by a model including as predictors the age of the user, the amount of time spent on Instagram, and the editing of the profile’s photo. The test on the significance of the predictors highlighted a negative link between both the amount of time spent on Instagram and the dependent variable (β = −0.319; t38 = −2.067; *p* < 0.05) and the age and the dependent variable (β = −0.283; t38 = −1.882; *p* < 0.05), indicating that the difference decreases with the increase in the amount of time spent on Instagram and the age.

A significant model was found having as a dependent variable the difference between the “self on Instagram” and the “others” and, as a predictor, the “editing of the posted photos” (F1, 38 = 5.224; *p* < 0.05, R2 = 0.121). The test of the predictors displayed a negative link between the two variables (β = −0.348; t38 = −2.286; *p* < 0.05) indicating a decrease of this distance for individuals who do not edit their posted photos. No significant models were found when the dependent variable was the difference between “self after like” and “others”. Having as a dependent variable the difference between the “self after no like” and “others”, a significant model was found (F2, 37 = 3.787; *p* < 0.05, R2 = 0.171) including the variables as predictors the “number of followers” and the “editing of the posted photos”. The test on the significance of the predictors highlighted a negative link between the dependent variable and both the number of followers (β = −0.379; t38 = −2.397; *p* < 0.05) and the editing of the posted photos (β = −0.324; t38 = −2.050; *p* < 0.05), indicating that the difference decreases with the increase in the number of followers and for individuals who do not edit the photos. Finally, a significant model having as predictors the “time spent on Instagram” and the “number of followers” was found when the dependent variable was the difference between “offline self” and “others” (F3, 36 = 5.342; *p* < 0.05, R2 = 0.308). The test on the predictors highlighted a positive link between the dependent variable and the time spent on Instagram (β = 0.607; t36 = 3.475; *p* < 0.05), while a negative link was observed with the number of followers (β = −0.546; t36 = −3.033; *p* < 0.05). This indicates that the difference increases with the amount of time spent on Instagram and the decrease of the number of followers. The results of the most significant regression models are reported in Figure 2.

## 4. Discussion

The aim of the study was to explore the influence of the use of a popular SNS, Instagram, on adolescents’ constructions of self and interpersonal experience. In line with our hypotheses, self-construction on Instagram and after receiving a “like” were less adherent to how adolescents think others see them than offline self-construction was. Moreover, change was higher when adolescents construed themselves on Instagram than when they construed themselves offline. These results suggest that the self-expression and self-disclosure favored by SNSs [8] lead adolescents to step away from their past self and from how they think others see them. They probably find the opportunity to selectively present themselves online and receive validation of these selective self-presentations. This is also thanks to the new connections allowed by SNSs with groups reflecting aspects of their identity that they wish to explore [55,56].

The influence of feedback on adolescents’ construction of self and their interpersonal experiences [50] was confirmed by the results of the present study. Specifically, there was no difference in adolescents’ self-acceptance and in social desirability when referring to “self” on Instagram in comparison to the “offline self”, but there was an increase in the same aspects after having received “likes” for their posts. On the other hand, there was a general decrease in the same indices after not receiving “likes”. This data is in line with previous research showing that adolescents do not portray their “ideal selves” through SNSs [28], but belongness and self-esteem needs are satisfied by receiving “likes” [67]. Previous studies on self-esteem found an association between the use of SNSs and low self-esteem [68], whereas online chatting with peers or strangers, or receiving support, were associated with positive self-esteem [57,58,59]. All these results seem to indicate that it is not so much the use of a SNS in itself that promotes an increase in self-esteem, but the receipt of positive feedback.

Another point of debate concerns social isolation. An extensive body of literature [8,69,70,71] showed the positive effect of the use of social networks and the Internet in decreasing social isolation and enhancing social connections. Pittman and Reich [72], in particular, indicated that Instagram and other image-based platforms, in contrast to text-based platforms (e.g., Twitter) or mixed platforms (e.g., Facebook), ameliorated loneliness due to the enhanced intimacy they offer. Nevertheless, evidence of a “rich-get-richer” phenomenon is provided, whereby young people whose offline friendship quality is perceived as being “high” had greater benefits from online communicative activities, whereas individuals with limited offline social networks do not develop quality friendships online, and may spend excessive amounts of time on SNSs, which can increase social isolation [34,73]. The results of the present study confirmed this increase in social isolation only when participants evaluated themselves after receiving no “likes”. These findings are consistent with previous results in suggesting that sharing photos and videos without receiving any positive feedback may lead to a greater sense of loneliness [30,61,62].

No gender differences were found in terms of any of the previous dimensions. Nevertheless, the regression models showed a decrease of self-acceptance on Instagram in the case of female adolescents, and in participants who edited photos. This data is consistent with the descriptive data that female adolescents are those who edit photos more often, and the results of a focus group study involving teens, which showed that girls were more likely to report expecting their close friends to like their posts on Facebook and Instagram, and asking for likes if their friends had not yet done so [74]. Barker [3] found a higher collective self-esteem in women. Jackson et al. [75] found that girls had lower physical appearance and athletic self-concepts, but higher behavioral self-concepts than boys. However, the measures used in both studies were independent of SNS use. Thus, more research is needed to further explore the moderating effect of gender on the relationship between SNS use and self-esteem.

The increase in usage frequency, the number of followers, and the editing of posted photos predicted the increase in social isolation on Instagram, thus suggesting that this effect is dependent on these aspects, but not on gender and age of the user. Previous studies [76,77] confirmed that these latter aspects might not predict an increase in social isolation. Instead, there is some previous evidence to suggest that spending excessive amounts of time on SNSs can lead to symptoms of depression, which can then increase the risk of social isolation [31].

A new type of data from the present study concerns social desirability because this dimension has not been explored in previous studies. Increased time spent on Instagram predicted an increase in social desirability after receiving a “like”, thus, suggesting that the more time people spend on Instagram, the higher is the effect in terms of feeling socially desirable. Finally, an increase in the number of followers predicts an increase in social isolation after receiving no “likes”, thus heightening the fact that having more followers does not decrease social isolation and may even increase the sense of loneliness derived by feeling oneself to be further away from others after receiving no “likes”. This data can be considered in the light of the previous literature [29,78,79,80] on the links between narcissism and the need for followers and approval; the more followers adolescents have, the more they depend on their approval for feeling less alone.

### Limitations

A limitation of the study was the limited number of participants. This was mainly due to the methodology used. Even if it allowed us to gather rich and complex data within a group, it was very time consuming. This aspect might be addressed in future research by enlarging the sample or introducing a control group of adolescents who do not use Instagram. This aim is difficult to achieve because it is rare that an adolescent does not use this very popular SNS, as international data highlighted [13,14,15]. The methodology used in the present study enabled this limitation to be overcome because it allowed us to compare how adolescents perceive themselves when they are online with how they perceive themselves offline. Thereby, this solution might be further explored by extending the sample. Another of the limitations of the present study was the specific geographic area where the study was conducted. Including participants from other parts of Italy and other national contexts might enable the investigation of potential differences. Finally, the limited number of participants prevented us from applying algorithms or creating a structural equation model combining all the variables under examination, which might be useful to infer rules and create a unique general model. Again, future studies might contribute to fill this gap.

## 5. Conclusions

The results of the present study highlighted that Instagram might represent a great chance of self-expression and change, but the strenuous search for confirmation expressed in the time spent on Instagram, the number of followers, and photo editing, predicts an increase in social isolation and a decrease of self-acceptance. Overcoming the contraposition between a positive versus negative evaluation of the impact of SNS use on adolescents, which characterized most of the previous studies [48], these findings highlight that SNSs may enhance both wellbeing or risks [28,81]. Moreover, this study adds to the existing literature by revealing that more than being on Instagram, it is positive feedback that has an impact on self-acceptance, social desirability, and social isolation. These findings are useful in understanding the constant need for approval and confirmation by followers which adolescents appear to require today and suggests possible intervention strategies that take into account this need in order to promote wellbeing when using SNSs.

Adopting the perspective of personal construct theory allowed us to explore adolescents’ self-construction and interpersonal experience by referring to their relevant dimensions rather than investigating the more general relationship between pre-defined personality traits and SNS use, as the majority of previous studies have done [82,83]. Furthermore, the chosen methodology offered an alternative to a longitudinal or between-group design and gave this study the possibility of investigating the relationship between the above-mentioned dimensions and the use of a specific SNS. Moreover, new dimensions (e.g., looking-glass self, social desirability, and change) were explored, which proved to offer a fruitful direction for future research, but further studies are needed to explore their relations with SNSs use. This is crucial in order to design future studies and intervention policies with regard to teenagers’ social networking practices. Instead of refusing or accepting a priori that SNSs are harmful or beneficial tools for adolescents’ wellbeing or indicating who can benefit from them or not depending on his/her personality traits, the present study suggests the key importance of identifying those features of SNS use that can enhance the potential of SNSs for adolescents’ psychosocial experience.

## Figures and Tables

**Figure 1 ijerph-17-06952-f001:**
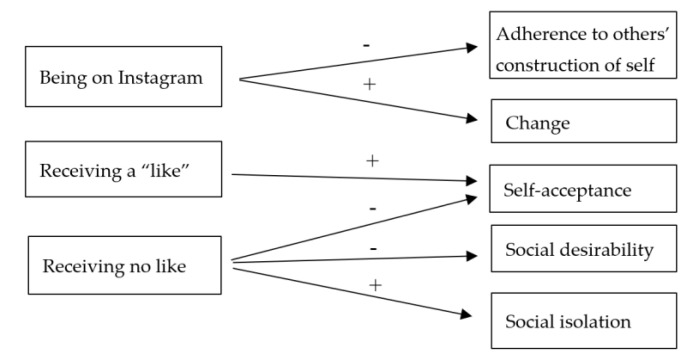
Diagram of the main results of ANOVA test.

**Figure 2 ijerph-17-06952-f002:**
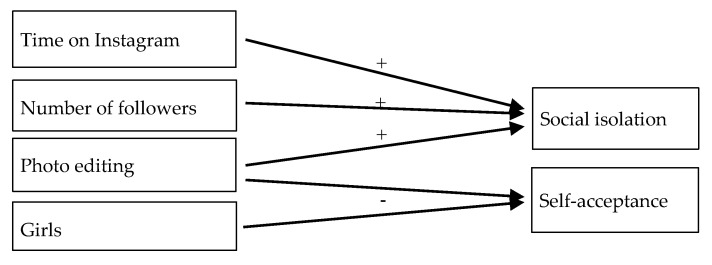
Diagram of the main results of regression analyses.

**Table 1 ijerph-17-06952-t001:** Main effects of the within factor for each 2 × 2 mixed ANOVA. For each model the mean of the two groups, the mean standard error, the F statistic value and the *p*-value are reported.

	Mean (MSE)	MSE	F (1,38)	*p*	Partial η^2^
Offline self–How others see me	0.498 (0.039)	0.040	18.417	<0.001 *	0.324
Self on Instagram–How others see me	0.689 (0.040)				
Offline self–How others see me	0.498 (0.039)	0.041	14.022	<0.001 *	0.270
Self after the like–How others see me	0.681 (0.042)				
Offline self–Past self	0.933 (0.041)	0.035	11.411	0.002 *	0.231
Self on Instagram–Past self	1.050 (0.035)				
Offline self–Ideal self	0.804 (0.036)	0.047	0.002	0.966	<0.001
Self on Instagram–Ideal self	0.798 (0.035)				
Offline self–Ideal Self	0.804 (0.036)	0.047	4.235	0.047 *	0.100
Self after the like–Ideal self	0.700 (0.031)				
Offline self–Ideal self	0.804 (0.036)	0.047	44.316	<0.001 *	0.538
Self after no like-Ideal self	1.131 (0.037)				
Offline self–Person I like	0.829 (0.040)	0.048	0.071	0.792	0.002
Self on Instagram–Person I like	0.820 (0.040)				
Offline self–Person I like	0.829 (0.040)	0.052	4.158	0.048 *	0.100
Self after the like–Person I like	0.718 (0.037)				
Offline self–Person I like	0.829 (0.040)	0.045	26.411	<0.001 *	0.410
Self after no like–Person I like	1.072 (0.039)				
Offline self-Others	0.835 (0.027)	0.030	0.028	0.867	0.001
Self on Instagram-Others	0.841 (0.027)				
Offline self–Others	0.837 (0.028)	0.050	2.353	0.133	0.058
Self after the like–Others	0.786 (0.028)				
Offline self–Others	0.784 (0.024)	0.031	57.741	<0.001 *	0.542
Self after no like–Others	0.962 (0.031)				

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
