# Peer review of "How Can Adolescents Benefit from the Use of Social Networks? The iGeneration on Instagram"

_ijerph, 2020, doi:10.3390/ijerph17196952_

Round 1

Reviewer 1 Report

The research is of great importance in the European context where most adolescents use SNS, which is evidenced by the results presented here. The influence of SNS in the construction of self is demonstrated in a small group of adolescents. Though with a small sample that prevents a discussion of public heatlh in a broader context, the paper is overall well-written and concise.

Major comments: 

I felt a gap between lines 78 to 79 that is explained from line 94. When the authors say "For this reason" to focus on Instagram, the reason is only clear two paragraphs latter. I would suggest to enhance this part of the introduction and to moving the "real" reason of studying Instagram before shaping the focus of the work.

The way that the focus (objective) and hypotesis are presented is confusing. I would suggest the authors to rewrite the final sentence of the introduction in order to make hypothesis and objective clear to the readers.

I understand that the participants are anonymous. However any additional information on the context and place in which the survey was designed could improve the manuscript and provide the readers a valuable information on the extent of the discussion and implications of the results. This is completely absent from the Materials and Methods section but, given the affiliations of the authors, I assume that the participants were Italians from Padua and Milan. With that said, I wonder whether the conclusions are valid for non-Italian adolescents or even for adolescents in other parts of the country (assuming that the survey was conducted in the North). In summary, the authors could explaining why these forty adolescents were chosen and where where they were from. 

Finally, the authors discuss their results by comparing with existing literature while claiming in the Conclusions that the investigation is "crucial in order to design future studies and intervention policies". Thus, I belive that the discussion could benefit with some deeper critical interpretation of the results, beyond the mere comparision with existing literature. E.g. Despite the well recognized limitations (, what are the possible policy implication of the results for the offline lives at school, home etc? Are SNS healthy for these interviewd teenagers? E.g. l. 340: did the authors detect any symptoms in their own research? Though not mandatory for acceptance, this surely improve the value of the manuscript for a broader readership.

Minor recommendations:

l79 could be rewritten to: For this reason, the current study will specifically focus on Instagram, a very popular...

l80 same: Since its beginnings in 2010, Instagram has...

l205 are the authors meaning that boys use Instagram for 50% less time than girls? Not clear.

Author Response

  1. I felt a gap between lines 78 to 79 that is explained from line 94. When the authors say "For this reason" to focus on Instagram, the reason is only clear two paragraphs latter. I would suggest to enhance this part of the introduction and to moving the "real" reason of studying Instagram before shaping the focus of the work.

Response 1: We have re-organized the introduction and moved the part on Instagram after presenting how teens use SNS. Now the aim of the study is more clearly linked to what is written before on the need of exploring specific aspects of SNS use in relation to some dimensions of adolescents’ identity and relationships.

  1. The way that the focus (objective) and hypotesis are presented is confusing. I would suggest the authors to rewrite the final sentence of the introduction in order to make hypothesis and objective clear to the readers.

Response 2: We have now presented the aim and the hypotheses as a separate paragraph (at the end of the introduction) from the presentation of the dimensions. We hope now it is clearer.

  1. I understand that the participants are anonymous. However any additional information on the context and place in which the survey was designed could improve the manuscript and provide the readers a valuable information on the extent of the discussion and implications of the results. This is completely absent from the Materials and Methods section but, given the affiliations of the authors, I assume that the participants were Italians from Padua and Milan. With that said, I wonder whether the conclusions are valid for non-Italian adolescents or even for adolescents in other parts of the country (assuming that the survey was conducted in the North). In summary, the authors could explaining why these forty adolescents were chosen and where where they were from. 

Response 3: We have added information on the context and the participants and a comment in the discussion regarding the geographical specificity.

  1. Finally, the authors discuss their results by comparing with existing literature while claiming in the Conclusions that the investigation is "crucial in order to design future studies and intervention policies". Thus, I belive that the discussion could benefit with some deeper critical interpretation of the results, beyond the mere comparision with existing literature. E.g. Despite the well recognized limitations (, what are the possible policy implication of the results for the offline lives at school, home etc? Are SNS healthy for these interviewd teenagers? E.g. l. 340: did the authors detect any symptoms in their own research? Though not mandatory for acceptance, this surely improve the value of the manuscript for a broader readership.

Response 4: In the conclusion we have better stated the meaningfulness of the results and the policy implications.

  1. Minor recommendations:

l79 could be rewritten to: For this reason, the current study will specifically focus on Instagram, a very popular...

l80 same: Since its beginnings in 2010, Instagram has...

l205 are the authors meaning that boys use Instagram for 50% less time than girls? Not clear.

Response 5: Changes have been done and l205 clarified.

Reviewer 2 Report

This study explored the influence of the use of Instagram on adolescents’ constructions of self and interpersonal experience. This could be valuable research in light of guiding future studies and  intervention policies in terms of teenagers’ social networking practices.

This article scientifically sounds, nevertheless, I have some concerns that I think they will improve your paper.

Abstract

Althought it is well structured, background information is missing.

Introduction

The introduction is exhaustive and provides a quite complete overview of the matter, but it seems too extensive. Some of the authors could be moved to the Discussion section. This section should have 3 main paragraphs:

The first paragraph mentions the questions and issues that outline the background of the study and establishes, the context and relevance of the problem.  

The second paragraph includes the importance of the problem and unclear issues.

The last paragraph briefly mentions the main aim of the work and highlights the main conclusions.

Some authors are misquoted, the year should be added.

Some important authors who have published on this topic (Vandenbosch, Eggemont...) are not cited.

Methods

Please include information about the period of the interviews. If they were taken during the quarantine, literature should also include social media use/effects during isolation.

Conclusions

In the Conclusions section, you should emphasize the main aspects and the novelty provided by this study. As it stands now, it’s difficult to see the novelty that the study brings, it seems just a new common sense insight.

I am looking forward to reading the final version of the manuscript.

Author Response

Abstract

Althought it is well structured, background information is missing.

- Background information has been added. 

Introduction

The introduction is exhaustive and provides a quite complete overview of the matter, but it seems too extensive. Some of the authors could be moved to the Discussion section. This section should have 3 main paragraphs:

The first paragraph mentions the questions and issues that outline the background of the study and establishes, the context and relevance of the problem.  

The second paragraph includes the importance of the problem and unclear issues.

The last paragraph briefly mentions the main aim of the work and highlights the main conclusions.

  • The introduction has been re-structured according to the suggestions of the reviewer.

Some authors are misquoted, the year should be added.

  • These corrections have been done.

Some important authors who have published on this topic (Vandenbosch, Eggemont...) are not cited.

  • These authors have been added.

Methods

Please include information about the period of the interviews. If they were taken during the quarantine, literature should also include social media use/effects during isolation.

  • Interviews were conducted before quarantine. More information about recruitment and sample has been added in the method section.

Conclusions

In the Conclusions section, you should emphasize the main aspects and the novelty provided by this study. As it stands now, it’s difficult to see the novelty that the study brings, it seems just a new common sense insight.

  • The main aspects and novelty of the study has been better underlined in the first paragraph of the conclusion. Other aspects are reported in the following paragraphs.

Reviewer 3 Report

Although the paper seems to study a topic of interest for the readers of the  IJERPH, it has several drawbacks that make it, as currently presented, inadmissible for a journal publication. This reviewer has identified the following main issues:

  • The abstract can be rewritten to be more meaningful. The authors should add more details about their final results in the abstract. The abstract should clarify what is precisely proposed (the technical contribution) and how the proposed approach is validated;
  • Introduce the chart for the given algorithm with description;
  • The authors should consider more recent research done in the field of their study (especially in the years 2019 and 2020 onwards);
  •  In the references in the Introduction section, the authors cite some works. However, they have not indicated the advantage or disadvantage and their relations to this paper. It’s a little confusing;
  • Please use a simple diagram or figure to illustrate the whole idea of this paper, and the modification it has been made from previous work or traditional framework.
  • Comparison with recent study and methods would be appreciated;
  • Need a detailed explanation of the preprocessing steps.
  • Clarify the finding Error rate and accuracy in the performance analysis section.
  • More extensive simulations and more figures are needed;
  • I suggest that you add some more results. Some more theoretical Math analysis, equations and a good mathematical model, some simulation results and some comparison of the presented scheme with other schemes. Maybe some figures for the simulation results or the comparisons;
  • The writing of the paper needs a lot of improvement in terms of grammar, spellings and presentations.

Author Response

The abstract can be rewritten to be more meaningful. The authors should add more details about their final results in the abstract. The abstract should clarify what is precisely proposed (the technical contribution) and how the proposed approach is validated;

Response: Something has been added regarding the proposed approach at the end of the abstract.

Introduce the chart for the given algorithm with description

Response: We have not added the chart because no algorithm was used in the study. IN the limitation section we have specified that this was not possible due to the small number of data collected.

The authors should consider more recent research done in the field of their study (especially in the years 2019 and 2020 onwards);

Response: More recent studies have been added.

In the references in the Introduction section, the authors cite some works. However, they have not indicated the advantage or disadvantage and their relations to this paper. It’s a little confusing.

Response: We have clarified the link between the references and our study and re-structured the introduction in order to make it clearer.

Please use a simple diagram or figure to illustrate the whole idea of this paper, and the modification it has been made from previous work or traditional framework.

Response: A figure has been added to illustrate the idea of the paper.

Comparison with recent study and methods would be appreciated

Response: Comparison with recent studies has been added in the discussion by referring to more recent studies and comparison with other methods has been discussed in the limitation and conclusion section.

Need a detailed explanation of the preprocessing steps.

Response: A detailed explanation of the preprocessing steps has been included in the new version of the manuscript.

Clarify the finding Error rate and accuracy in the performance analysis section.

Response: We are sorry, but we do not really understand what the reviewer is asking here. We would kindly ask to the reviewer to clarify the request.

More extensive simulations and more figures are needed;

Response: We are sorry again, but we do not really understand what the reviewer intends by "more extensive simulations". This is an empirical study and no simulations have been carried out. Furthermore, we would kindly ask the reviewer to suggest what figures should be added so that we can address the point

I suggest that you add some more results. Some more theoretical Math analysis, equations and a good mathematical model, some simulation results and some comparison of the presented scheme with other schemes. Maybe some figures for the simulation results or the comparisons;

Response: One again we do not succeed in understanding what the reviewer is asking. What does the reviewer mean by "theoretical Math analysis", "equations" and "good mathematical model"?

The writing of the paper needs a lot of improvement in terms of grammar, spellings and presentations.

Response: We thank the reviewer for this suggestion. The paper was revised by a professional editing service (Academicproofreading.com) in order to improve grammar, spelling and readability.

Reviewer 4 Report

This paper discusses the influence of the use of a popular social networking site on adolescents’ constructions of self and interpersonal experience. The manuscript is well written, however, it is not well organized. A section on related work should be added for purposes of comparison and better observation of the study main contributions. There is an absurd amount of references. Remove unnecessary and outdated references. If possible, present the socio-economic data of the participants.

Author Response

Thank you very much for the encouraging comments.

Round 2

Reviewer 3 Report

The authors presented the requested corrections, however, in the opinion of this reviewer, they could have presented a cover letter punctuated for each requested correction. I am in favour of publication.

Author Response

Thank you for your final comments. We have further clarified the presentation of the results by adding an additional figure representing the results of the regression analyses and moving the figure representing the results of ANOVA test in the result section.